# Quantification and Phenotypic Characterization of Extracellular Vesicles from Patients with Acute Myeloid and B-Cell Lymphoblastic Leukemia

**DOI:** 10.3390/cancers14010056

**Published:** 2021-12-23

**Authors:** Marijana Miljkovic-Licina, Nicolas Arraud, Aicha Dorra Zahra, Patricia Ropraz, Thomas Matthes

**Affiliations:** 1Laboratory for R&D in Hematology, Center for Translational Research in Onco-Hematology, University of Geneva Medical School, 1206 Geneva, Switzerland; marijana.licina@unige.ch (M.M.-L.); aicha.zahra@ulb.be (A.D.Z.); patricia.ropraz@unige.ch (P.R.); 2Department of Oncology, Hematology Service, Geneva University Hospitals, 1205 Geneva, Switzerland; 3Department of Diagnostics, Clinical Pathology Service, Geneva University Hospitals, 1205 Geneva, Switzerland; arraud@hp-hl.de

**Keywords:** extracellular vesicles, flow cytometry, acute leukemia, AML, B-ALL

## Abstract

**Simple Summary:**

Extracellular vesicles (EVs) are lipid bound vesicles secreted by cells into the extracellular space. They play an important role in cell-to-cell communication by transporting diverse messenger molecules, including DNA, RNA, lipids, and proteins. Cancer cells in patients with acute leukemia also produce EVs that can influence cells from the surrounding bone marrow microenvironment and play a role in disease progression. In our study, we applied several methods such as cryo-electron microscopy, fluorescence triggering flow cytometry, and nanoparticle tracking analysis to quantify and phenotypically characterize EVs from a series of acute leukemia patient blood samples. Our results show how these methods can be used to study the functional role of EVs in leukemia development and to evaluate their potential as targets for therapy and as biomarkers of the disease.

**Abstract:**

Extracellular vesicles (EVs) act in cell-to-cell communication, delivering cargo from donor to recipient cells and modulating their physiological condition. EVs secreted by leukemic blasts in patients with leukemia have been shown to influence the fate of recipient cells in the bone marrow microenvironment. Methods to quantify and to characterize them phenotypically are therefore urgently needed to study their functional role in leukemia development and to evaluate their potential as targets for therapy. We have used cryo-electron microscopy to study morphology and size of leukemic EVs, and nanoparticle tracking analysis and fluorescence triggering flow cytometry to quantify EVs in platelet-free plasma from a small cohort of leukemia patients and healthy blood donors. Additional studies with a capture bead-based assay allowed us to establish phenotypic signatures of leukemic EVs from 17 AML and 3 B-ALL patients by evaluating the expression of 37 surface antigens. In addition to tetraspanins and lineage-specific markers we found several adhesion molecules (CD29, and CD146) to be highly expressed by EVs from B-ALL and several leukemic stem cell antigens (CD44, CD105, CD133, and SSEA-4) to be expressed by EVs from AML patients. Further improvements in analytical methods to study EVs are needed before potentially using them as biomarkers for leukemia prognosis and follow-up.

## 1. Introduction

Acute leukemias (AL) are a heterogenous group of aggressive hematologic malignancies characterized by the proliferation of either myeloid (AML) or lymphoid (ALL) progenitor cells called leukemic blasts in the bone marrow (BM). Special niches, which are composed of mesenchymal stem cells, adipocytes, macrophages, osteoblasts, and other stromal cell types constitute the microenvironment in which leukemic blasts proliferate. The complex interplay between these different cells which determines leukemic cell survival and proliferation depends on direct cell-to-cell contact and intercellular communicators such as cytokines, chemokines, growth factors, adhesion, and extracellular matrix molecules. A crucial role in the dialogue between blasts and niche cells has been recently attributed also to extracellular vesicles (EVs), sub-micrometric particles delimitated by a phospholipid bilayer [1].

EVs are classified according to their size and mode of biogenesis into: microvesicles (50–1000 nm) originating by budding from plasma membranes, apoptotic bodies (50–5000 nm) deriving from cells undergoing programmed cell death, and exosomes (30–150 nm) developing from early endosomes released into the extracellular space after fusion with the cell membrane [2,3].

The study of EVs is challenging due mainly to their small size of a few hundred nanometers, their limited surface area, and reduced number of proteins available for antibody binding. To isolate EVs, various methodologies have been implemented including ultracentrifugation, size exclusion chromatography, microfluidics, immunoaffinity capture-based techniques, precipitation, and more [4]. Guidelines have been published by the International Society for Extracellular Vesicles to implement standardization in EV analysis and reporting (MISEV 2018; MIFlowCyt-EV 2020) [5,6].

EVs have been shown to contain biologically active molecules that can be transferred from cell to cell [7]. This mode of communication has the capacity to deliver a diverse array of messages from EV-producing to EV-recipient cells, since EVs can harbor bioactive molecules such as DNA, coding, and non-coding RNAs, regulatory miRNAs, lipids, and proteins [8]. In particular, miRNAs have been shown to be transported by tumor-derived EVs in various hematologic diseases and to modulate gene expression in surrounding target cells, as in CLL [9] and multiple myeloma [10,11].

The group of Whiteside was among the first to study EVs from AL patients [12,13]. They isolated CD34^+^ EVs from the plasma of patients and showed that by adding these EVs to cultures of NK cells, NKG2D expression was decreased and NK function impaired. Other studies showed that leukemia-derived EVs can target BM stroma cells leading to increased production of pro-inflammatory cytokines and reduced stromal cell-derived factor 1 (SDF1) mRNA expression [14]. Leukemia-derived EVs were found to suppress normal residual hematopoietic progenitors either indirectly by reprogramming of stromal cells or directly via suppressive miRNAs that target c-myb, a transcription factor involved in hematopoietic progenitor differentiation and proliferation [15]. On the one hand, it has been suggested that leukemic EVs are participating in the transformation of the BM niche into a cancer favorable microenvironment [16]. On the other hand, EVs derived from BM niche cells were also found to influence normal and leukemic hematopoietic stem cell (HSC) behavior. Thus, EVs produced by stromal cells from AML patients were found to contribute to chemoresistance of AML blasts against tyrosine kinase pathway inhibitors [17]. Moreover, EVs produced by embryonic stem cells enhanced HSC cell survival and proliferation, and upregulated the expression of several early HSC markers in these cells [18].

In this study, we analyzed EVs from the plasma of patients with AL using cryo-electron microscopy, nanoparticle tracking analysis (NTA), flow cytometry, and flow cytometry bead-based assays, in accordance with the published guidelines regarding EV research. We showed how these methods allow the morphologic description, quantification and immunophenotyping of EVs, and—as a proof of principle—that several antigens, detected by flow cytometry on AML blasts, are also found on corresponding EVs, such as CD29, CD44, or SSEA-4. Further development of these approaches is needed to help unravel the role of EVs in the pathophysiology of acute leukemias.

## 2. Materials and Methods

Detailed methods can be found in the Appendix A.

### 2.1. Leukemia Patient Samples, Cell Isolation and Primary Cell Culture

Peripheral blood (PB) from healthy blood donors (HBDs) and from AL patients were obtained after informed consent. The study was approved by the Medical Ethics Committee of the Canton Geneva (study no. 08-001 and 2020-00176). Diagnosis and clinical and laboratory characteristics of leukemia patients are detailed in Appendix A. CD34^+^ cells were enriched using the UltraPure CD34 Microbead kit (Miltenyi Biotec, Bergisch Gladbach, Germany, #130-100-453) and magnetic-activated cell-sorting separation columns (Miltenyi Biotec, #130-042-201). CD34^+^ cell viability and purity were assessed by flow cytometry after staining cells with CD34-PE-Vio770 (Miltenyi Biotec, clone: AC136, #130-113-180), CD45-VioGreen (Miltenyi Biotec, clone: REA747, 130-110-638), and 7-Amino-Actinomycin D (7-AAD) viability dye for live/dead cell discrimination (Beckman Coulter, Brea, California, CA, USA, #B88526). Samples were analyzed on a Navios 10-color flow cytometer (Beckman Coulter, Brea, CA, USA) and the data interpreted with the Kaluza software (Analysis Version 2.1, Beckman Coulter). CD34^+^ cells were subsequently cultured in StemSpan^TM^ SFEM II serum-free medium (STEMCELL Technologies, Vancouver, BC, Canada, #09605), supplemented with Flt3L, IL-3, IL-6, SCF and TPO (StemSpan^TM^ CD34^+^ expansion supplement; STEMCELL Technologies, #02691).

### 2.2. Isolation and Purification of EVs from Platelet-Free Plasma

All the experiments in this study were performed according to the guidelines published by the International Society of Extracellular Vesicles to implement standardization in EV analysis and reporting [5,6].

After centrifugation of PB samples (2× 2000× *g* for 15 min) platelet-free plasma (PFP) was recovered and frozen at −80 °C until further analysis. For isolation of EVs from PFP samples a total exosome precipitation reagent was used (Life Technologies, Invitrogen, Carlsbad, CA, USA). Precipitated EVs were then recovered by centrifugation at 10,000× *g* for 5 min at RT, resuspended in 100–500 μL of phosphate-buffered saline (PBS) and kept at −80 °C for long-term storage. Total protein was quantified in PFP-derived EV preparations by the Micro BCA™ Protein Assay kit (Thermo Fisher Scientific, Waltham, MA, USA).

### 2.3. Nanoparticle Tracking Analysis (NTA)

To determine the concentration and size distribution of EVs in the purified samples, NTA was performed using a ZetaView TWIN^®^ apparatus (Particle Metrix GmbH, Inning am Ammersee, Germany). Standard EV preparations from plasma of healthy individuals or conditioned media (CM) from the K562 cell line (HansaBiomed Life Sciences, Tallinn, Estonia) were used as controls. Between each sample passage, the counting chamber was thoroughly flushed with 0.1 µm filtered PBS. EV samples were diluted in 0.1 µm filtered PBS to an appropriate concentration before analysis and videos of the light-refracting particles were recorded with the following settings: 25 °C fixed temperature, 11 positions, 1 cycle, sensitivity 80, shutter 100, min. brightness: 30; min. area: 10; max. area: 1000, 3 measurements per sample. After capture, the videos were analyzed by the in-build ZetaView Software 8.05.10 (Particle Metrix GmbH, Inning am Ammersee, Germany).

### 2.4. Fluorescence Triggering Flow Cytometry (FT-FCM)

EVs were analyzed on a Navios flow cytometer (Beckman Coulter, Brea, CA, USA) as previously described [19,20]. Briefly, fluorescence settings were tested daily using Flow-Check Pro Fluorospheres (Beckman Coulter, #A63493). Prior to labeling, antibody solutions were centrifuged at 18,000× *g* for 5–10 min in order to eliminate potential protein aggregates. For labeling with CD34-PE or CD81-PE antibodies (Beckman Coulter, #A07776, and Miltenyi Biotech, #130-118-481, respectively), 15 μL PFP was diluted to 100 μL with 0.1 μm-filtered Cell Wash Buffer (CWB, BD) containing 2% BSA and incubated overnight with 5 μL of the pure antibody, at room temperature in the dark. Samples were then diluted to 500 μL with CWB for FT-FCM analysis. As negative control for antibody labeling, PFP samples were prepared as described above, with CWB supplemented with 0.1% Triton X-100 (AxonLab, Stuttgart, Germany). As reagents-only control, samples were prepared with CWB containing 2% BSA and either CD34-PE or CD81-PE antibody. Prior to analysis, one μm fluorescent beads (Ultra Rainbow beads, Spherotec GmbH, Lake Forest, IL, USA) were added to each sample at a final concentration of 5 × 10^4^/μL, both as an internal reference for relative size and acquired volume. Each sample was then measured using FL-2 triggering signals for the detection of EVs labeled with PE-conjugated antibodies.

### 2.5. Multiplex Bead-Based EV Flow Cytometry Assay

EV preparations were subjected to bead-based multiplex EV analysis by flow cytometry (MACSPlex Exosome Kit, Miltenyi Biotec, Bergisch Gladbach, Germany), as previously described [21,22]. Briefly, samples were diluted with 0.1 μm filtered MACSPlex buffer (MPB) supplemented with 2% BSA to a final concentration of 50 μg protein/μL in low-protein binding test tubes (Axygen, Union City, CA, USA). Subsequently, capture beads, containing thirty-nine different antibody-coated bead subsets, were added to each sample, and incubated overnight. After one washing step EVs bound to beads were either incubated for 2 h with a mix of APC-conjugated CD9, CD63, and CD81 antibodies (5 µL each, MACSPlex kit; Miltenyi Biotec) or with CD34-APC antibody (5 µL, Beckman Coulter, #A07776). After another washing step EV/bead complexes were resuspended in PBS and analyzed on the flow cytometer. Between 11,000 and 14,000 beads were analyzed and median fluorescence intensities (MFI) for all capture bead subsets were recorded after subtraction of respective MFI values from matched buffer or serum-free media controls. Two negative control bead subsets were included in all analyses: beads coated with an isotype mouse IgG1 and beads coated with a recombinant isotype control antibody against keyhole limpet hemocyanin (REA control).

### 2.6. Statistical Analysis

For comparison of two means, Student’s *t* test (two-sided, paired) was used, and for multiple mean comparisons, one-way or two-way variance analysis (ANOVA) followed by Bonferroni post hoc test. GraphPad Prism was used for statistical calculations and presentations (GraphPad Software, San Diego, CA, USA). Results were considered statistically significant at *p* < 0.05.

## 3. Results

### 3.1. Characterization of Morphology, Size, and Counts of Leukemia-Derived EVs by Cryo-EM and NTA

Detailed characterizations of EVs from plasma have been performed previously via Cryo-EM [23,24]. However, EVs extracted from plasma of leukemia patients have been less well characterized. We used PFP from HBDs and leukemic patients who had been extensively phenotyped during the routine work-up for our study (Appendix A). To avoid the formation of aggregates often obtained when high-speed centrifugation of PFP samples is used, we enriched EVs with a polymer-based precipitation method [25]. Isolated EVs were then visualized with Cryo-EM yielding information on their size and morphology, such as membrane structures and vesicle content. In total, images of 50 particles were analyzed from a leukemic patient sample (Figure 1A,B and Appendix A) and 50 particles from a sample of a HBD.

Typical EVs, limited by a lipid bilayer of 4 nm, constituted >95% of the visualized particles, with over 50% of them single, spherical vesicles (156 ± 98 nm), with a translucent interior or with an electron dense cargo (128.5 ± 53 nm) (Figure 1A,B). Thirty % of the vesicles contained either two bilayer membranes (132 ± 40 nm) or were multilayered (260 ± 107 nm) (Appendix A), formed by vesicle membrane reorganization during the preparation process [26,27].

Figure 1C show the size distribution of all analyzed EVs, with 50% of them being between 100 and 200 nm. No difference was found when we compared the mean particle size from PFP samples of HBDs and leukemic patients by NTA (Figure 1D). Appendix A show typical examples for the different patient cohorts. Since size distribution between microvesicles and exosomes can overlap significantly [28] our preparations might contain a mixture of both types of EVs, although with a majority of exosomes.

The NTA method allows determining particle counts in a liquid solution. NTA measurements showed a range of 10^10^–10^11^ particles in the PFP samples analyzed, with a tendency for higher counts in AML patients without reaching statistical significance and significantly higher counts in B-ALL patients compared to HBDs (Figure 1E).

Taken together, Cryo-EM and NTAs showed that the EV preparations contained typical EVs with no morphology or size differences between healthy controls and AL patients but quantitative differences in samples from B-ALL patients compared to HBDs. These preparations were used for further analysis.

### 3.2. Quantification and Immunophenotyping of Leukemia-Derived EVs by Fluorescence Triggering Flow Cytometry

As results from NTA can be confounded by the presence of membrane fragments, protein aggregates or other contaminating nanoparticles in the sample preparation, we used fluorescence triggering flow cytometry (FT-FCM) as a method to quantify EVs based on EV surface marker expression [19,20]. This method is based on triggering the detection of EVs on a fluorescent signal instead of the traditional light scattering triggering used in flow cytometry and NTA, and thus is more specific and sensitive than these two methods, respectively.

We tested this approach using tetraspanin CD81 as a pan-vesicular marker and CD34 as a marker for leukemia-derived EVs (8). Reagents-only and Triton X-100-treated samples were used as negative controls. We first established titration curves to determine the range of PFP sample volumes that could be used for measurements (Appendix A). Typical results obtained with FT-FCM for EVs from a HBD and an AML patient are shown in Figure 2A; results for all samples labelled with the CD34 and CD81 antibodies in Figure 2B, respectively. CD34^+^ EV concentrations were significantly, 3-fold higher in CD34^+^ AML patient samples (range: 10–80 × 10^3^ EVs/µL PFP) compared to either the healthy individuals, CD34^−^ AML, or B-ALL CD34^−^ patient samples (range: 3–10 × 10^3^ EVs/µL PFP) (Figure 2B, left panel). Staining of samples with CD81-PE antibody showed no significant differences (Figure 2B, right panel).

The concentration of CD34^+^ EVs from CD34^+^ AML patients correlated with the number of total white blood cell (WBC), as well as with CD34^+^ leukemic blast cell counts (Figure 2C,D, left panels, respectively), whereas no correlation was observed for CD81^+^ EVs counts (Figure 2C,D, right panels, respectively).

Taken together, our data demonstrate that labelling of leukemia-derived EVs with CD34 antibody and analysis with FT-FCM allows to quantify CD34^+^ EVs from peripheral blood of leukemia patients.

### 3.3. Determination of Phenotypic Profiles of Leukemia-Derived EVs

In order to study the phenotypic profile of EVs from AL patients in more detail we used a multiplex bead-based flow cytometry assay, which allows the simultaneous, semi-quantitative detection of 37 different antigens (Figure 3A,B). Several of the antigens present in the assay are of relevance for leukemic blast phenotyping.

We first validated this approach by studying EVs produced in vitro by Kasumi-1, a CD34^+^ AML cell line. Culture supernatants were recovered after 24 h from Kasumi-1 cells seeded at three different concentrations, EVs were isolated and used first for the evaluation of the input amounts allowing an optimal read-out of the assay (11,000–14,000 events of counted beads; Figure 3C,D). For subsequent experiments input amounts of 50 µg EVs or 5 × 10^8^ EVs/assay were used. This corresponded to 2 × 10^6^ Kasumi-1 cells at culture start (Figure 3D). After incubation with a mix of beads coupled to one of thirty-nine different antibodies, the EVs bound to the beads were detected by an APC-conjugated pan-tetraspanin antibody mix (CD9/CD63/CD81). Figure 3E shows representative results for eight antigens. Analysis of Kasumi-1 cell surface antigens by conventional flow cytometry confirmed presence of the CD9, CD63, and CD81 antigens also on the surface of the parental leukemic cells (Appendix A, upper panels).

We than compared detection of all 39 antigens on Kasumi CM-derived EVs either by the pan-tetraspanin antibody mix or by the anti-CD34 antibody (Figure 4). Six antigens were found to be abundantly present on EVs bound to beads and revealed with the CD9/CD63/CD81 antibody mix: the tetraspanins CD63 and CD81, two integrins CD29 and CD49e, and two leukemic cell-associated surface antigens CD44 and CD105 (Figure 4B). Revelation with the CD34 antibody confirmed these data (Figure 4B). Analysis of Kasumi-1 cell surface antigens by conventional flow cytometry confirmed presence of the three tetraspanins, several integrins, and three leukemic cell markers on the parental leukemic cells, e.g., CD34, CD44, and CD105 (Appendix A).

Similar phenotypic profiles were observed with other AML-derived cell lines, MV-114 and OCI-AML3 (Appendix A). These results indicate that EVs derived from AML cell lines carry at least some of the surface markers expressed by the parental cells. Similarly, we detected CD19, CD20, and HLA-DR on EVs derived from JeKo-1 and BJAB, two B cell lines; and CD2 on EVs derived from Jurkat cells, a T cell line (Appendix A). Interestingly, EVs from Jurkat cells showed only high expression of CD2, but not of CD3 and CD4; although, both markers are expressed on Jurkat cells (data not shown).

After validation of the multiplex assay on cell lines, we then set out to analyze EVs from PFP samples of leukemia patients and HBD (Figure 5A,B). The antigenic profiles of EVs from HBDs showed, in addition to pan-vesicular tetraspanins and integrins, the highest signal intensities for antigens related to platelets: CD41b, CD42a, and CD62p (Figure 5C, upper panel). Other antigens from blood cell populations were only weakly detected on EVs, such as T cell antigens (CD2, CD3, CD4, and CD8). When we further analyzed EV antigen profiles in samples from AML patients, we detected expression of CD133 and CD44, two hematopoietic stem/progenitor cell markers, as well as the type I transmembrane protein CD105 in samples where the leukemic blasts expressed these antigens (Figure 5C, middle panel). In AML samples where the blasts were CD133 negative or in B-ALL samples, CD133 was not detected among the antigens expressed on EVs (data not shown and Figure 5C, lower panel). Furthermore, high expression of CD19 and HLA-DR was found on EVs derived from B-ALL samples (Figure 5C, lower panel).

Comparison between all the samples from the three different patient groups (7 HBDs, 17 AML, and 3 B-ALL patients) showed differential expression for several antigens (Figure 6). The expression of the tetraspanins CD9, CD63, and CD81 was particularly high in B-ALL samples (Figure 6A), as were the expression of B-cell antigens CD19, CD24, and HLA-DR (Figure 6B), and cell adhesion molecules CD29 and CD146 (Figure 6C). EVs from AML samples expressed also higher amounts of CD63 and CD29 (Figure 6A,C, respectively), compared to samples from HBDs.

Expression of the leukemic stem cell antigen CD44 was increased in EVs from both leukemia types, AML and B-ALL, compared to EVs from HBDs (Figure 6D). CD105 was found significantly increased in B-ALL (3/3 samples) and CD133 in 6/17 AML samples (Figure 6D). Of interest, the glycolipid antigen SSEA-4 was found increased in 3/17 AML samples (Figure 6D). Standard flow cytometry analysis of SSEA-4 expression on the three AML samples confirmed the presence of this glycolipid on the surface of the parental leukemic cells (Appendix A). No significant difference in the EV surface expression of platelet proteins (CD41b, CD42a, and CD62p) was found between the groups (Figure 6E and data not shown).

Taken together, these data show homogeneous overexpression of a number of antigens on EVs from B-ALL patients, as well as overexpression of other antigens on AML samples, although overexpression was there more heterogenous and patient-specific.

For one AML patient with EVs overexpressing CD133 and CD44 we were able to analyze follow-up samples during and after chemotherapy (Figure 7A). At day 5 of induction therapy, circulating leukemia-derived EVs were still detected; at day 10 of treatment onwards, the patient showed profound pancytopenia and the CD133 and CD44 signatures as well as the signatures of EVs derived from other blood cell populations such as platelets became undetectable by capture beads immunoassay (Figure 7B). These findings were in concordance with quantification of CD34^+^ EVs by FT-FCM (Figure 7C).

Standard flow cytometry performed in parallel on peripheral blood and BM samples at day 10 showed disappearance of blasts in peripheral blood (sensitivity of 10^−3^), but not in BM (still 100% infiltration). Disappearance of BM blasts and negative minimal residual disease (MRD) were observed only at day 30, just before start of the second cycle of induction chemotherapy. It seems therefore not possible to use disappearance of leukemia-derived EV leukemic markers as detected by capture beads assay from peripheral blood as a surrogate marker for MRD detection.

## 4. Discussion

EVs constitute a way of intercellular communication different from direct cell-to-cell contact or soluble factors such as cytokines. By virtue of their virus-like size, cell-derived EVs can transport information in the form of proteins or genomic material to cellular targets in the microenvironment or, via the bloodstream, to far away tissues and organs [29]. In order to study their role in tissue homeostasis and disease, robust laboratory methods to quantify and characterize them are therefore mandatory.

Flow cytometry together with bead-based assays have emerged as methods, which allow quantification of EVs and analysis of membrane surface proteins and signatures, similar to studies on whole cells. Based on these two assays we have analyzed PFP samples from AL patients and compared them to samples from HBD.

FT-FCM has been used previously to quantify EVs in plasma from HBD and values of ca. 5 × 10^8^ EVs/mL have been reported, based on expression of Annexin 5, CD41, and CD235a [20]. We used the tetraspanin CD81 as a pan EV marker and found ca. 10 times less, 1.4 × 10^7^ EVs/mL. One explanation for our lower values could be decreased sensitivity of CD81 detection due to few molecules present on EV surfaces or presence in plasma of EVs not expressing CD81, as observed by others [30,31]. EV concentrations in plasma from AL patients showed increased values, in some patients up to a 100-fold, compared to HBD. It is not clear whether this increase is solely due to increased amounts of EVs secreted by the leukemic blasts or those secreted by inflammatory cells, activated platelets, or endothelial cells.

CD34 is highly expressed by leukemic blasts in most cases. When we used a CD34 antibody to stain EVs we found concentrations of CD34^+^ EVs of ca. 10^7^/mL in HBD. Most probably, these EVs are secreted by vascular endothelial cells and derived from platelets, both cell types expressing CD34 on their surface [32,33]. In AL patients CD34^+^ EVs were found increased 5–10 times compared to HBD but only in patients with CD34^+^ blasts, not in patients where the blasts were CD34^−^. Numbers of CD34^+^ blasts correlated to leukemic blast counts, and we showed that during chemotherapy EV and blast counts decreased in parallel. Unfortunately, the high background of CD34^+^ EVs in HBD does not allow the use of this approach for sensitive MRD detection. In the situation where blasts are cleared from peripheral blood but still detectable by flow cytometry in BM, CD34^+^ EV numbers in peripheral blood have already dropped to background values. The same probably holds true for many other markers that are expressed by tumor cells but also by their normal counterparts. In fact, as in standard flow cytometry, it seems rather impossible to use only one marker for the quantification of leukemic EVs, and at least double or triple staining would be required.

To establish a phenotypic signature of leukemia-derived EVs we have used a bead-based assay developed by Wiklander et al. [21]. This assay allows the detection and semi-quantification of EVs expressing 37 different proteins. Unfortunately, several important proteins such as CD34, CD123, or CD33, which are present on most AML blasts, are not part of this commercial assay.

In AML samples EVs expressing CD29, CD44, or SSEA-4 were found to be consistently increased compared to HBD, in subgroups of AML patients CD133 (6/17), CD105 (7/17), CD146 (4/17) were also found to be increased. In B-ALL EVs expressing CD19, CD24, CD29, CD44, CD105, CD146, and HLA-DR were increased compared to HBD. EVs expressing the tetraspanins CD9, CD63, and CD81 were also increased in B-ALL compared to AML patients and HBD. We confirmed, although not for all, that these antigens are also expressed on the membrane of the leukemic cells. For several other antigens that are expressed on leukemic cells like CD2, CD3, or CD4 on the Jurkat cells, only CD2 was found on EVs. Similarly, CD20 expressed on the surface of the JeKo-1 B cell lymphoma line was not found on EVs. Others have reported similar observations. Miguet et al. observed in a proteomic study of isolated EVs that CD3, CD5, and CD8 were not present on EVs derived from T cells, and that CD19 was not found on EVs derived from B cells [34]. Belov et al. have used an antibody microarray to compare the surface protein profiles of CLL cells with those of their EVs [31]. They found that EVs expressed only a subset of ca. 40% of the proteins detected on CLL cells from the same patients. Several typical CLL associated proteins like CD5, CD19, CD31, and CD44 were expressed by EVs, others like CD21, CD49c, and CD63 were expressed at low levels, and yet CD20 and CD23 were not expressed at all. None of these proteins was detected on EVs from the plasma of age- and gender-matched HBD.

As detection depends on antibody affinity and antigen density on EV membranes these divergent results could simply reflect different sensitivities of the detection methods used. Alternatively, it has been speculated that the repartition mechanism of antigens into the membrane of EVs could be selective and different for each antigen. Some antigens clustered in lipid rafts on the cell membrane could be transferred collectively to EV membranes. Further studies are clearly needed to clarify this important issue.

Several of the antigens found to be expressed by leukemia-derived EVs correspond to proteins with functions in adhesion.

CD146 has been described to be highly expressed on blasts in ca. 30% of B-ALL patients [35]. CD146 functions as a receptor for laminin alpha 4, a matrix molecule that is broadly expressed within the vascular wall. The presence of CD146 on B-ALL EVs could play a role in their adhesion to endothelium, and blocking this interaction could inhibit EV binding and phagocytosis by endothelial cells.

CD44 is another adhesion molecule that is expressed on hematopoietic cells and has been implicated in the interactions between BM stromal layers and hematopoietic progenitors. Variants of CD44 are expressed on blasts from AML patients and several of them have been associated with poor prognosis [36]. The presence of CD44 on EVs secreted by leukemic blasts could increase binding of EVs to stromal cells and facilitate their uptake.

CD105 expression was described in a subset of AML patients [37], and high CD105 expression was found to significantly correlate with poor overall and progression-free survival [38]. Whether CD105 on leukemic EVs plays any role in their fate and in the pathophysiology of AML patients is currently unknown. Among antigens increased in EVs from AML patients we found in 3/17 cases a high expression of stage-specific embryonic antigen, SSEA-4. SSEAs constitute a family of glycosphingolipids expressed on human embryonic stem cells in a tightly controlled stage-specific manner during embryonic development [39,40,41]. More recent studies have shown that these antigens are also expressed in adults by a subpopulation of mesenchymal stem cells (MSC) [42]. Expression on leukemic blasts has not been described so far, but presence on blasts as well as EVs could make SSEA-4 an interesting new target for AML diagnosis and treatment.

Our study shows that—as a proof of principle—EVs and antigens expressed by them could potentially be used as new biomarkers for AML and ALL. To validate the clinical and therapeutic utility of these biomarkers, studies on larger cohorts of patients as well as follow-up studies of patients at relapse are needed. Designing multiplex bead assays specifically for the phenotyping of leukemia-derived EVs including antigens such as CD34 and CD123, as well as frequently expressed aberrant markers, would allow the establishment of comprehensive phenotypic profiles. For the development of such assays the screening of antibody clones suited for EV detection targeting antigens with a high density to allow EV binding to beads would be mandatory to achieve sufficient high sensitivity of detection. In addition, assays applicable for the clinical practice are needed, with a short turnaround time and reliable quantifications, such as a recently published flow cytometry-based test that has been used to measure platelet-derived EVs in unmanipulated fresh blood [43]. Informative investigations, such as comparisons between leukemia-derived EVs in diagnostic and relapse situations, would thus become possible and could lead to the development of strategies to block EVs reaching target cells.

## 5. Conclusions

Extracellular vesicles (EVs) play an important role in cell-to-cell communication by transporting diverse messenger molecules. In patients with acute leukemia, cancer cells produce EVs that can influence cells from the surrounding bone marrow microenvironment and play a role in disease progression.

In our study, we quantified and characterized EVs phenotypically from a series of acute leukemia patients and from healthy blood donors. We found several adhesion molecules (CD29, and CD146) to be highly expressed by EVs from B-ALL and several leukemic stem cell antigens (CD44, CD105, CD133, and SSEA-4) to be specifically expressed by EVs from AML patients. These antigens could constitute potential new targets for therapy and new biomarkers of the disease. Additional studies involving a larger cohort of patients are needed to validate clinical and therapeutic utility of the herein described biomarkers.

## Figures and Tables

**Figure 1 cancers-14-00056-f001:**
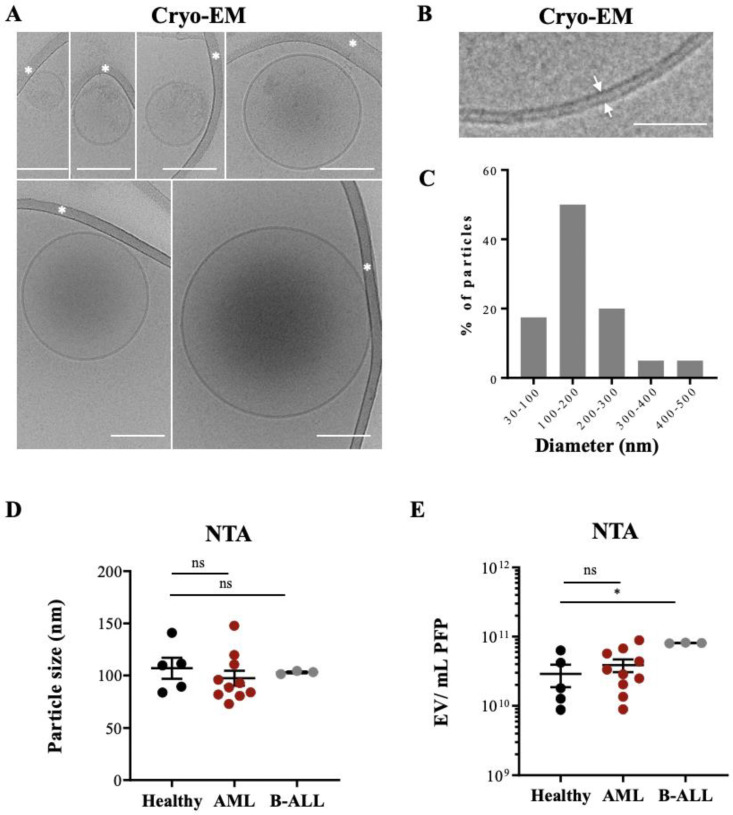
Cryo-EM of EVs isolated from leukemia-derived PFP samples: (**A**) Assembly of six micrographs showing single EVs of different sizes embedded in a thin film of frozen EV preparation. Some vesicles show an electron dense cargo in their lumen. Scale bar, 100 nm. White asterisks mark strands of the supporting carbon net. (**B**) The lipid bilayer at the vesicle membrane is resolved in two discrete lines 4 nm apart (arrows). Scale bar = 50 nm. (**C**) The histogram shows the size distribution of 50 EVs from one AML patient, determined by Cryo-EM. (**D**) PFP samples isolated from HBD (n = 5), AML (n = 10), and B-ALL (n = 3) patients were analyzed by NTA in light scatter mode and median particle size was determined. (**E**) Particle concentration (EVs/mL) of PFP samples from HBD and AL patients were measured by NTA. One-way ANOVA with post hoc Bonferroni was performed for multiple group comparison. * *p* < 0.05; ns, non-significant.

**Figure 2 cancers-14-00056-f002:**
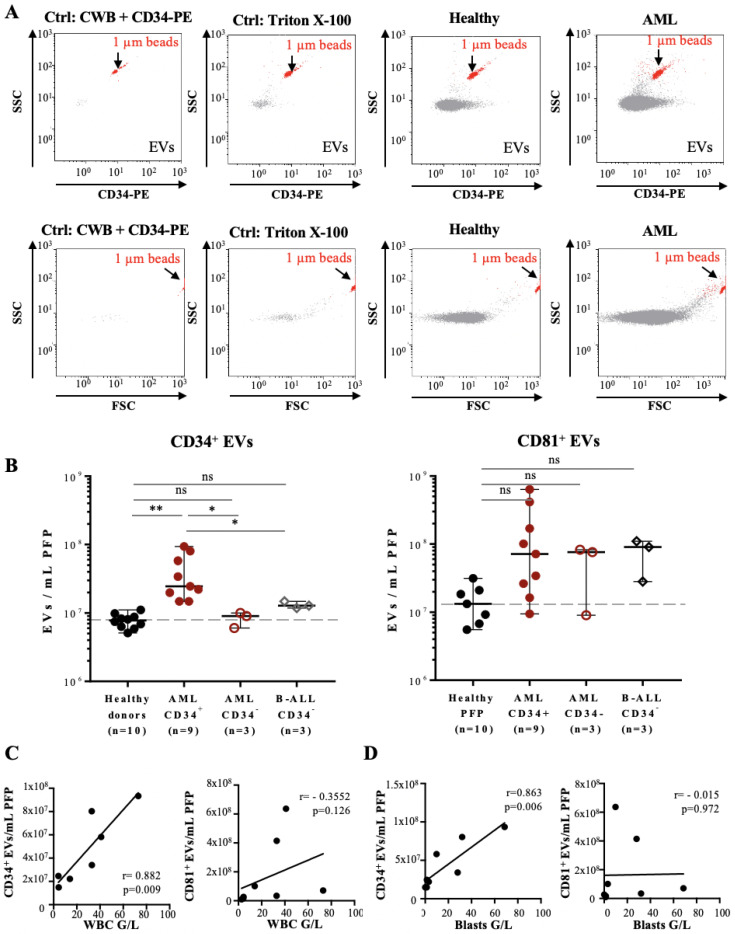
Flow cytometry analysis of PFP samples from leukemia patients based on the fluorescence triggering method: (**A**) Representative FL-2 vs. SSC (upper panels) and FSC vs. SSC (lower panels) dot plots of PFP sample analysis by fluorescence triggering: reagents-only control (CWB containing 2% BSA and CD34-PE), negative control (PFP treated with Triton X-100, second panel), HBD (third panel), and AML patient (forth panel). PFP samples were incubated with CD34-PE antibody and FL-2 fluorescence used as a trigger, so that all detected events correspond to CD34^+^ EVs (upper panels). Here, 1 µm calibration beads (red) are used as an internal size and volume reference and are indicated in each dot plot. (**B**) Quantification of CD34^+^ (left panel) or CD81+ EVs (right panel) in PFP from HBDs (n = 10), CD34^+^ AML (n = 9), CD34-AML (n = 3), and CD34-B-ALL patients (n = 3). Error bars represent ± SEM (n = 3–10). One-way ANOVA with post hoc Bonferroni was performed for multiple group comparison. * *p* < 0.05; ** *p* < 0.01, and ns—non-significant. (**C**) Scatter plots with linear regression lines showing the correlation between the concentration of CD34^+^ EVs (left panel) and CD81^+^ EVs (right panel) per mL of PFP and the total WBC count (G/L) of seven AML patients at the time of diagnosis. r = 0.882, *p* = 0.009 (left panel), and r = −0.355, *p* = 0.126 (right panel). (**D**) Scatter plots with linear regression lines showing the correlation between the concentration of CD34^+^ EVs (left panel) and CD81^+^ EVs (right panel) per mL of PFP and the leukemic blast count (G/L) of seven AML patients at the time of diagnosis, respectively. r = 863, *p* = 0.006 (left panel), and r = −0.015, *p* = 0.972 (right panel).

**Figure 3 cancers-14-00056-f003:**
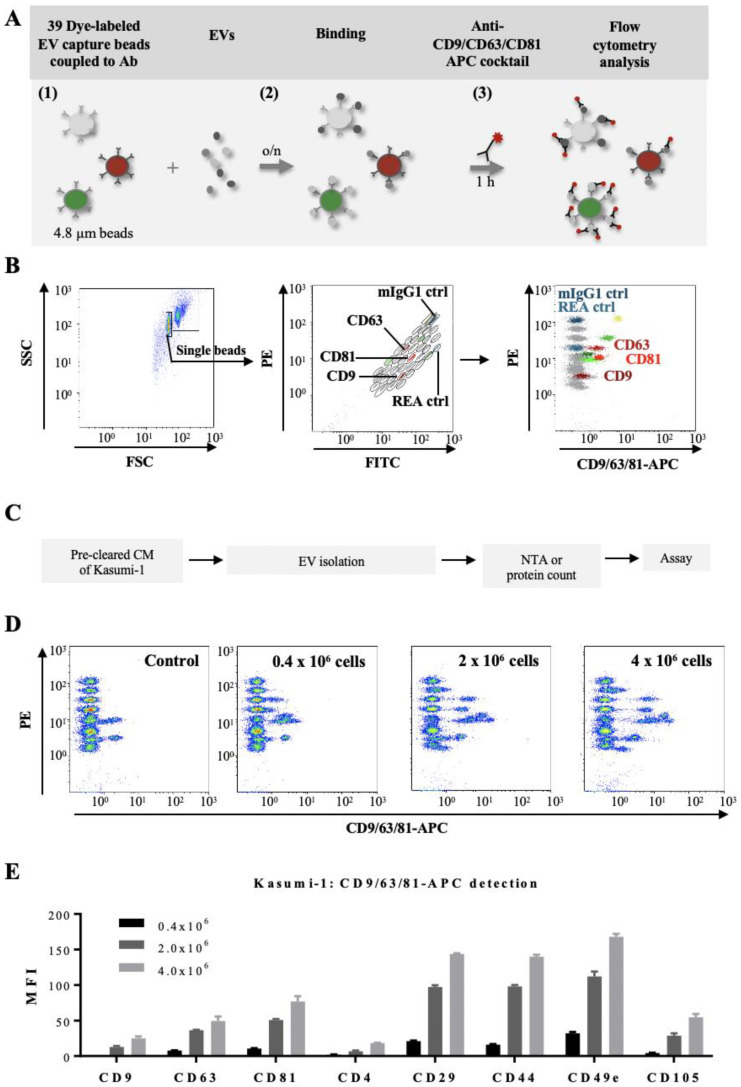
Multiplex bead-based flow cytometry assay: simultaneous detection of 39 different antigens on EVs: (**A**) Schematic representation of the multiplex bead assay, based on a cocktail of 39 fluorescently-labelled bead populations (MACSPlex Exosome Capture beads, 4.8 µm) that can be distinguished due to their differential expression of FITC and PE fluorescence (1). Each bead is coupled to an antibody directed against a different antigen. After overnight incubation of the EV preparation with the bead mixture (2), EVs bound to a specific bead can be counterstained with an APC-conjugated antibody or an antibody mix, and then analyzed by flow cytometry, based on the fluorescence characteristics of both capture beads and APC-conjugated antibodies (3). (**B**) Illustration of the gating strategy for exclusion of doublets (left panel), discrimination of differently fluorescently labelled bead populations (middle panel), and measurements of APC fluorescent intensities of single bead populations (right panel). Positive bead populations are highlighted in colors. Grey populations correspond to beads that did not bind EVs or to beads binding EVs that are not detected by the APC-conjugated antibody cocktail (anti-CD9/CD63/CD81-APC). Two isotype control beads, one coupled to mouse IgG1 (dark blue) and one to an antibody recognizing keyhole limpet hemocyanin (REA control; light blue) were used as negative controls of non-specific binding. (**C**) Experimental outline of multiplex bead-based detection of EV surface signatures of Kasumi-1 cell line. Conditioned media (CM) of Kasumi-1 cells grown in a serum-free medium for 24 h were pre-cleared and used for EV isolation by the precipitation-based method from the cell culture supernatants (Invitrogen). The concentration of Kasumi 1-derived EVs was quantified by either total protein count or NTA and 50 µg or 5 × 10^8^ EVs, respectively, were used as assay inputs. (**D**) Three representative dot plots are shown for a range of 0.4 × 10^6^–4 × 10^6^ Kasumi cells at culture start, compared to serum-free medium control values (Control). (**E**) Median APC fluorescence intensities (MFI) for eight representative bead populations revealed with the CD9/CD63/CD81 antibody mix, after background correction (serum-free media values subtracted from measured Kasumi-1 CM-derived values). Shown are MFI values for three different input cell numbers.

**Figure 4 cancers-14-00056-f004:**
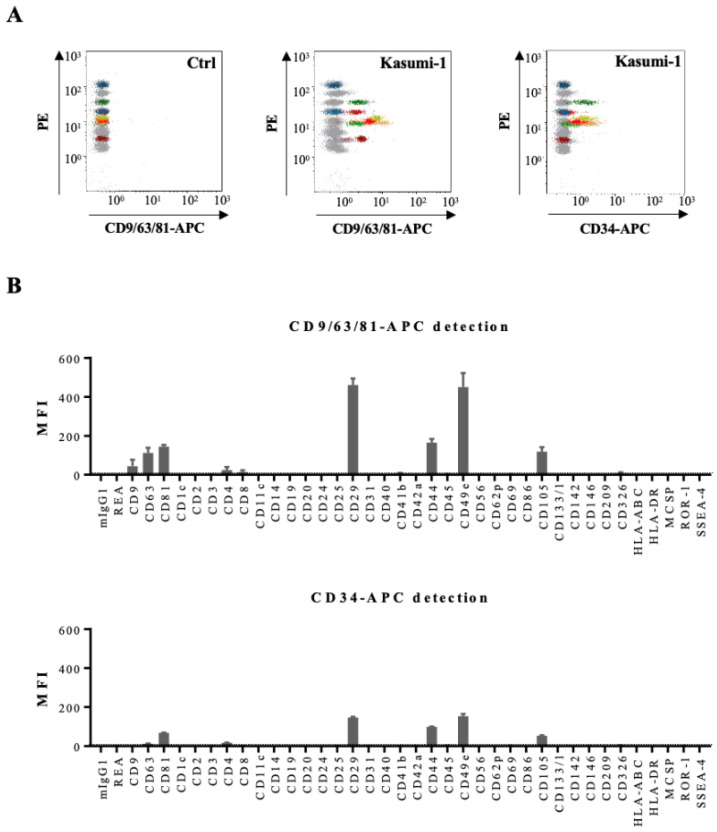
Multiplex bead-based flow cytometry assay: analysis of Kasumi-1 cell line. (**A**) Representative dot plots of multiplex bead-based flow cytometry analysis of Kasumi-1 CM-derived EVs showing measurements of single intensities of respective APC-stained bead populations, either anti-CD9/CD63/CD81-APC-stained (middle panel) or CD34-APC-stained (right panel) compared to the respective serum-free medium control values (Ctrl, left panel). (**B**) Median APC fluorescence intensities (MFI) for all 39 bead populations after background correction (serum-free medium values subtracted from measured Kasumi-1 CM values) with CD9/CD63/CD81-APC or CD34-APC detection.

**Figure 5 cancers-14-00056-f005:**
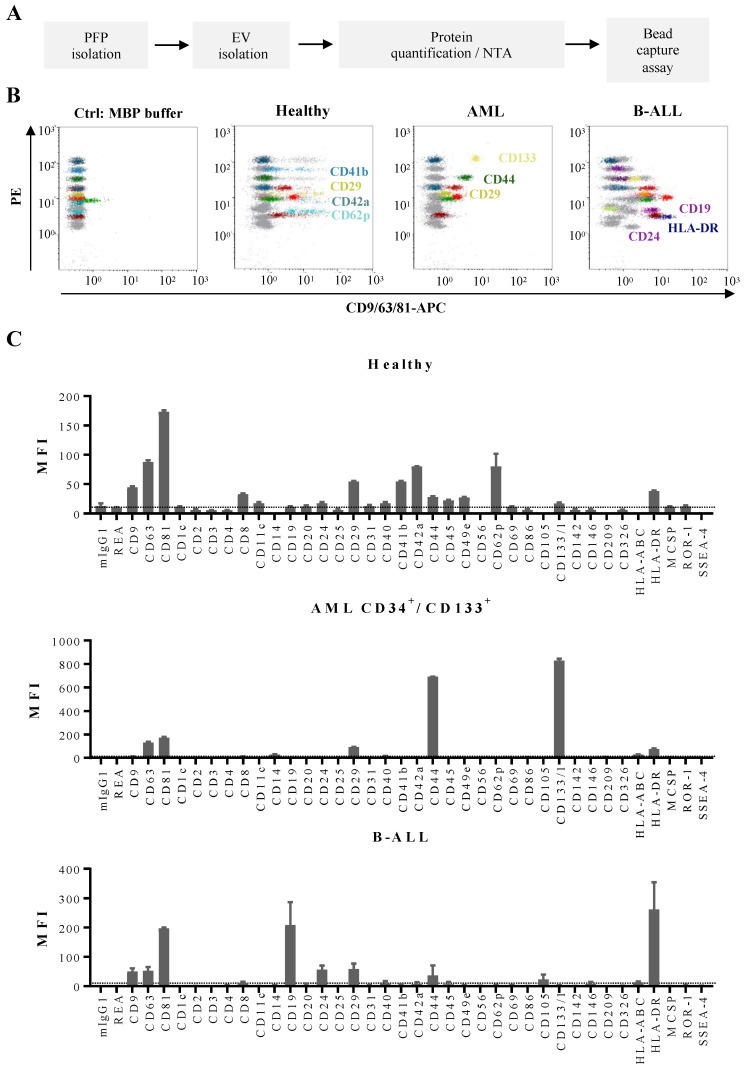
Multiplex bead-based flow cytometry assay: antigen profiling of EVs from PFP samples from leukemia patients at diagnosis and HBDs: (**A**) Experimental outline of multiplex bead-based detection of EV antigen profiles. The concentration of PFP-derived EVs was determined prior to use in the assay by either total protein count or NTA and defined doses of 50 µg or 5 × 10^8^ EVs, respectively, were used as assay inputs. (**B**) Representative dot plots showing the negative control (MBP buffer), as well as a sample from a HBD, an AML and a B-ALL patient. (**C**) Median APC fluorescence intensities (MFI) for all 39 bead populations after background correction (MBP buffer values subtracted from measured PFP values). Mean ± SEM of the three independent measurements (replicates) are shown (n = 1). For improved comparability of plots with different axis scaling, we included an arbitrary dotted line at an MFI value of 10.

**Figure 6 cancers-14-00056-f006:**
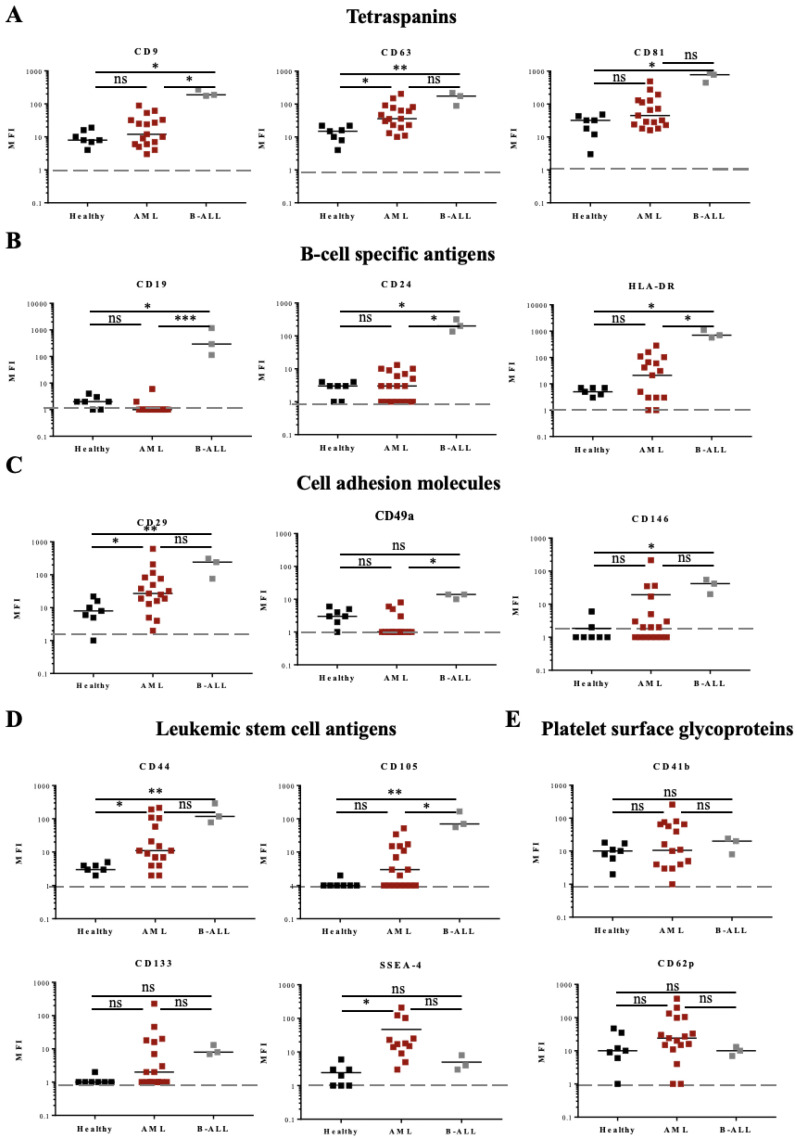
Fifteen antigen expressions from HBD, AML, and B-ALL patients by multiplex bead-based flow cytometry assay. Expression of 15 different antigens: (**A**) three tetraspanins, (**B**) three B-cell specific antigens, (**C**) three cell adhesion molecules, (**D**) four leukemic stem/progenitor cell antigens, and (**E**) two platelet glycoproteins was analyzed in PFP samples from AL patients (17 AML and 3 B-ALL) and compared to samples from 7 HBDs (**A**–**E**). One-way ANOVA with post hoc Bonferroni was performed for multiple group comparison. * *p* ≤ 0.05; ** *p* ≤ 0.01; *** *p* ≤ 0.001; ns—non-significant.

**Figure 7 cancers-14-00056-f007:**
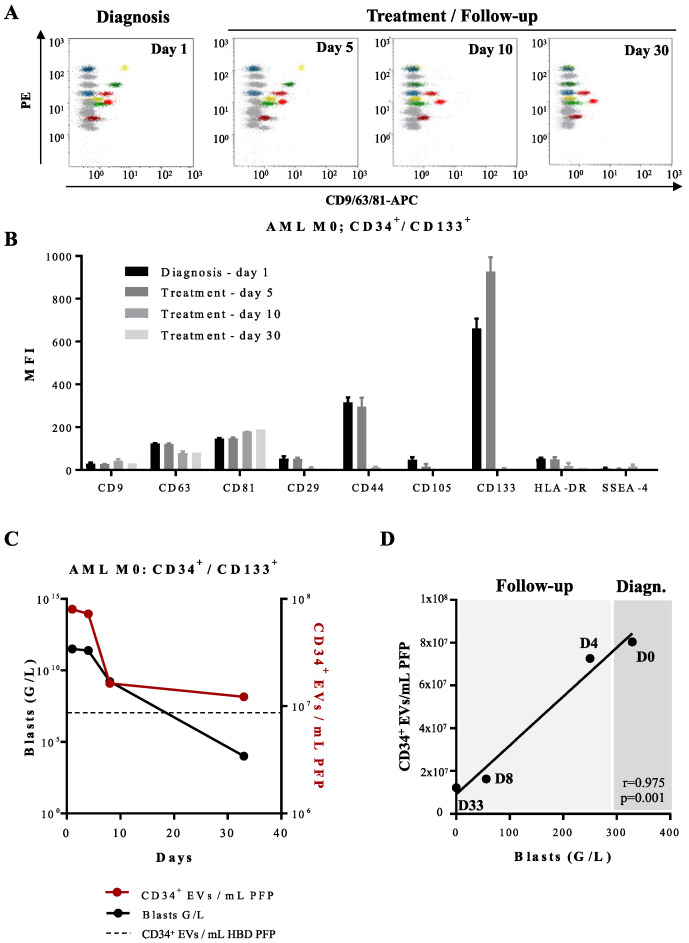
Multiplex bead-based flow cytometry assay—analysis of the antigenic profile of EVs obtained from an AML patient at diagnosis and during induction treatment: (**A**) Representative dot plots of multiplex analysis of PFP-derived EVs from an AML patient (Patient 1; Appendix A) at diagnosis (day 1), and during induction treatment on days 5, 10, and 30. (**B**) Median APC fluorescence intensities (MFI) for several bead populations revealed with the CD9/CD63/CD81 antibody mix at diagnosis (day 1), during the induction treatment on days 5, 10, and 30. EVs expressing CD44 and CD133 disappear from the phenotypic profile after day 5 of treatment. (**C**) Quantification of CD34^+^ EVs/mL of AML PFP measured by FT-FCM (red line) and leukemic CD34^+^ blasts (black line) measured by standard flow cytometry. Dotted line represents quantification of CD34^+^ EVs/mL of HBD PFP measured by FT-FCM. (**D**) Scatter plot with linear regression line showing the correlation between the concentration of CD34^+^ EVs/mL of PFP and the total blast count (G/L) of one AML patient (Patient 1; Appendix A) at the time of diagnosis, day 0 (D0), day 4 (D4), day 8 (D8), and day 33 (D33) of follow-up. r = 0.975, *p* = 0.001.

## Data Availability

The data presented in this study are available in this article and Appendix A.

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
