# Peer review of "Quantification and Phenotypic Characterization of Extracellular Vesicles from Patients with Acute Myeloid and B-Cell Lymphoblastic Leukemia"

_cancers, 2021, doi:10.3390/cancers14010056_

Round 1

Reviewer 1 Report

The research paper describes an impressive work made by Miljkovic-Licina and collaborators. Extracellular vesicles were isolated from cell culture supernatants and human-derived samples, and different biomarkers – able to distinguish between healthy individuals and leukemia patients – were identified.

The work is well written and organized, the experiments are well planned and fully described with appropriate negative controls. This work represents an important contribution in the field of EV-related biomarkers for liquid biopsy and therapeutics. I would recommend to clarify, at the end of conclusion section, that additional findings, involving a larger cohort of patients are needed to validate clinical and therapeutic utility of herein described biomarkers.

I only recommend a minor revision:

  • Line 189: check the number of the image S3C, I was not able to find it in the supporting files.

Author Response

We thank the reviewer's generous and positive remarks.

As suggested, in the revised version of the manuscript, we added a sentence at the end of the conclusion, as follows: “Additional studies involving a larger cohort of patients are needed to validate clinical and therapeutic utility of herein described biomarkers" (lines 596-597).

Line 189: We apologize for this omission. Indeed, the image S3C does not exist and the data described herein are related to Figures S3A and B. We have now modified the text corresponding to line 288 in the revised version of the manuscript accordingly.

Reviewer 2 Report

The study proposed by Dr. Matthes and colleagues is focused on the quantification and phenotypic characterization of extracellular vesicles (EV) from plasma of patients with acute leukemia (AML and B-ALL). EVs analysis still represents one of the major issues to be solved in order to translate their use in clinical settings. In this manuscript the authors applied different methods: cryo-electron microscopy, fluorescence triggering flow cytometry, nanoparticle tracking analysis (to study size, morphology and to quantify the EVs) and finally a capture bead-based assay to establish phenotypic signatures of EVs. The authors addressed several aspect concerning the combination of these methodologies highlighting the clinical utility and the future prospective.

This paper is interesting and technically sounds good. English language and style are fine. The research design is appropriated and the methodologies applied well described.

The conclusions are supported by the evidence presented and no additional experiments are required. However it was a pity that several of the antigen use in the bead-based flow cytometry assay don’t  perfect matched with leukemic disease and other more relevant (CD34 first of all) were not included in the kit: more information concerning the phenotypic profiles would have been obtained.

Finally, an appropriate use of statistics analysis is performed.

The overall subject is well written, however, in the introduction session, a more extensive overview of previous work would be advantageous.  For example, please consider to include the review written by Pando and colleagues (Extracellular vesicles in leukemia. Leuk Res. 2018 ) or by  Ratajczak MZ, Ratajczak J. (Extracellular microvesicles/exosomes: discovery, disbelief, acceptance, and the future? Leukemia. 2020).

Please also consider the corrections suggested below.

- In the “Materials and methods” session and in the “Supplementary materials and methods” session there are a lot of redundant information and often the same phrases are present in both the sessions. Please adjust this issue eliminating the redundant parts. Moreover, the references cited in the main text are different if compared to those listed in the “supplementary information” in the same phrase reported in both the session. Please check which are the correct ones.

- At page 3 in the 2.3 session are cited the K562 cell line. Could you please also insert this cell line in the paragraph “cell lines” in the “supplementary information”?

- In the “Results” session at page 10 at line 306 it is reported the analysis of 17 AML patients. The same in the legend of the Figure number 6, at page 12 line 320 and in the discussion session.  I had understood that the patients enrolled for this group were 12. What did I miss?

- Could you speculate something about why in the AML patients you have achieved overexpression of leukemic stem cell antigen more heterogeneous and patient-specific compare to B-ALL patients?

Minor Typing errors:

- Please insert in the first line of the manuscript the acronym “AL” for “Acute leukemia” instead of at line 67.

- Please insert in the first line at page 3 of the manuscript the explanation of the acronym “HBDs” that I suppose to be “Healthy Blood Donors” is it correct?

- Please insert in the last line at page 12 of the manuscript the explanation of the acronym “MRD” that I suppose to be “minimal residual disease” isn’t it?

- Please correct the word “suppementary” in the title “Supplementary material and methods”

Author Response

  • The overall subject is well written, however, in the introduction session, a more extensive overview of previous work would be advantageous.  For example, please consider to include the review written by Pando and colleagues (Extracellular vesicles in leukemia. Leuk. Res. 2018 ) or by  Ratajczak MZ, Ratajczak J. (Extracellular microvesicles/exosomes: discovery, disbelief, acceptance, and the future? Leukemia. 2020).

We thank the reviewer for his constructive and positive comments.

In the revised version of the manuscript, we have now added a more extensive overview of the previous work in the Introduction section, which includes, among others, the two suggested references: Pando et al, 2018 and Ratajczak MZ &Ratajczak, J. 2020. Please see the Introduction section in the revised version of the manuscript (lines 54 and 84).

  • Please also consider the corrections suggested below.

- In the “Materials and methods” session and in the “Supplementary materials and methods” session there is a lot of redundant information and often the same phrases are present in both the sessions. Please adjust this issue eliminating the redundant parts. Moreover, the references cited in the main text are different if compared to those listed in the “supplementary information” in the same phrase reported in both the session. Please check which are the correct ones.

We apologize for this redundancy. We solved this issue by eliminating the redundant parts of the text from the “Supplementary material and methods”. The “Material and methods” section of the manuscript is now more detailed and complete. The Supplementary material and methods section contains now only materials and methods that are not described in the main text of the manuscript.

The references cited in the main text are the correct ones. By removing redundant parts of the text from the “Supplementary material and methods” this issue is solved.

- On page 3 in the 2.3 session are cited the K562 cell line. Could you please also insert this cell line in the paragraph “cell lines” in the “supplementary information”?

We now inserted basic information of K562 cell line in the paragraph “cell lines” of the “Supplementary information, as follows: “K562 (obtained from ATCC-American Type Culture Collection) were cultured in RPMI-1640 containing Glutamax-1 (Invitrogen), supplemented with 10% FBS.”

- In the “Results” session on page 10 at line 306 it is reported the analysis of 17 AML patients. The same is in the legend of Figure 6, on page 12 line 320, and in the discussion session.  I had understood that the patients enrolled for this group were 12. What did I miss?

We apologize for these inconsistencies. There was a mistake in the Abstract section where a total of 12 AML patients were mentioned. Indeed, a total of 17 AML patients were used in the multiplex-bead-based flow cytometry assay as described in the results (Figure 6) and discussion section, as well as in Table S1. Only 12 AML patients (Table S1, samples 1-12) were used for the fluorescence triggering experiments (Figure 2) as we did not have enough PFP samples of the last 5 AML patients (Table S1, samples 13-17) to perform both tests.

- Could you speculate something about why in the AML patients you have achieved overexpression of leukemic stem cell antigen more heterogeneous and patient-specific compare to B-ALL patients?

One of the explanations of the heterogeneous results from the AML group is certainly the small sample size (n=17) and the heterogeneous AML subtypes with their different cytogenetic abnormalities (Table S2). A much larger panel of AML samples needs to be studied. In our very small B-ALL group (n=3), two patients were Phi neg and thus very similar in their cytogenetics. Finally, these results are in concordance with the observed phenotypes of the blasts cells analyzed by classical flow cytometry that are more heterogeneous in AML blasts than in B-ALL blasts.

Minor Typing errors:

- Please insert in the first line of the manuscript the acronym “AL” for “Acute leukemia” instead of at line 67.

We inserted the acronym “AL” for “Acute leukemia” as suggested by the reviewer in line 44 of the revised manuscript.

- Please insert in the first line at page 3 of the manuscript the explanation of the acronym “HBDs” that I suppose to be “Healthy Blood Donors” is it correct?

We inserted the explanation of the acronym “HBDs” e.g. healthy blood donors in the first line on page 3 that correspond to line 142 of the revised manuscript, as suggested by the reviewer.

- Please insert in the last line at page 12 of the manuscript the explanation of the acronym “MRD” that I suppose to be “minimal residual disease” isn’t it?

We inserted in line 453 on page 13 of the revised manuscript the explanation of the acronym “MRD” standing for “minimal residual disease” as suggested by the reviewer.

- Please correct the word “suppementary” in the title “Supplementary material and methods”

We apologize for this typing error. We corrected the word in the title of the revised version of the “Supplementary material and methods” section.

Reviewer 3 Report

In this manuscript Miljkovic-Licina et al., characterized  the extracellular vesicles (EVs) isolated  from the platelets-free-plasma of patients with acute leukemia (AL) either myeloid (AML) or lymphocytic (ALL) groups (n=15) by  using different approaches, accordingly to ISEV guidelines. EVs  are secreted by normal as well as cancer cells; the latter, by delivering their cargo,  can influence the fate of recipient cells in the bone marrow microenvironment, transforming them in cancer cells too. The authors reported that EVs  from both AL patient groups did not differ from those from healthy control (HC, n=10) regarding to the size, while EV counts were significantly higher in B-ALL  patients in comparison to HC. Interestingly, the authors identified a phenotypic signature distinguishing B-ALL patients from AML ones, by showing that some stem cell-derived surface antigens are expressed only by EVs from B-ALL patients.

The authors conclude that these antigens may represent potentially new targets for therapy and/or new biomarkers of AL.

Noteworthy, the identification of new AL biomarkers may contribute to a better understanding of the molecular bases of this disease, and may be useful in screening, diagnosis, prognosis, and monitoring as well as in predicting response to treatment. This makes the results of this research potentially outstanding, but even if the manuscript is well written, I have several concerns, including major comments, that need to be addressed.

Majors

- I recommend to increase the sample size. For discovery purpose, a sample size of n=50 cases and n=50 control are  required to have a high chance of detecting a (true) biomarker with a reasonable combination of attributes, expressed in a moderate fraction of cases (>30%) and separated by >- 2 SDs in proximal fluids (doi:10.1021/pr400132j).

-When the authors show the results obtained from fluorescence triggering flow cytometry, they explain the gating strategy used without showing it. They should include it in figure 2.

-The authors should include the “reagents-only controls” in the fluorescence triggering flow cytometry experiment to strengthen their data. They should prepare the sample using the same protocol but without adding the EVs. this allow to identify the possible background in the FL2 channel derived from the reagent used. This should be performed for both anti-CD34 and anti-CD81.

- The authors are not explaining the meaning and (if any) relevance of the increased levels of EVs found in B-ALL patients. As commented by the authors, these EVs can be released by other inflammatory cells. Did they check for the expression of CD31 or CD41a markers? Please comment on that.

- In Supplemental table 1 the authors are reporting the patient’s characteristics (citogenomics or mutations); did the authors evaluate any correlation with other clinical parameters? Please show.

Minors

  • The authors are referring to several methods to identify EVs in body fluids, but they are also mentioning that results can be confounded by the presence of membrane fragments, aggregates or other contaminating nanoparticles due to the sample preparation (i.e from blood to plasma). (lines 179-180). I will suggest to refer, among the different methods mentioned, to a very recent one applying flow cytometry on unmanipulated blood by discriminating integer EVs form any aggregates or destroyed EVs ( org/10.3390/cells10010085).
  • Line 41-43: logic structure of the sentence is not working (no verb?)
  • Line 269 missing the full stop (.) at the end of the sentence.
  • Lines 287-288 : logic structure of the sentence is not working (no verb?); alternatively, the “;” should be a “,”
  • Line 67; the acronym “AL” is present for the first time; the authors should write the entire name.
  • Line 71; the acronym “SDF1” is present for the first time; the authors should write the entire name.
  • Line 89; the acronym “HBDs” is present for the first time; the authors should write the entire name.
  • Line 115; the authors should not start a sentence using the numerals, but they should spell out it (“One” instead of “1”).
  • The sentence starting with the number “11’000” in the line 122, should be rephrased to avoid starting the sentence with numbers.
  • In figure 1D and 1E there are not present the error bars, the authors should add them.
  • In figure 1C, the authors should remove the red line below the word “diameter”.
  • Lane 159; the authors should not start a sentence using the numerals, but they should spell out it (“thirty” instead of “30”).
  • In supplementary figure 2, the authors should remove the red line below the words describing the axes.
  • Supplementary figure 4, the authors should remove the red line below the “t” in the title of the upper panel
  • Figure 3 panel B, in the left panel of the figure is present a typo, “FSS” should be substituted by “FSC”.
  • Supplementary figure 6A, in the third panel is present the “PE” word in the middle of the dot plot.
  • In the figure 5C is present in all the three histograms is present a dotted line that is not described in the figure legend, the authors should add the explanation.
  • Figure 6A, in the third plot the statistic bars overlap the figure. The authors should correct properly.
  • In figure 7C is present a dotted line that is not described in the figure legend, the authors should add the explanation.
  • In the lane 380 is present a double space at the beginning of the sentence “numbers of CD34+ blasts ….”.
  • Line 440, the acronym “MSC” is present for the first time; the authors should write the entire name.

Reviewer 4 Report

The authors presented a manuscript in which they provide new data on extracellular vesicles derived from acute leukemias, lymphoid and myeloid forms, identifying a high expression of adhesion molecules and antigens that could provide important implications in elucidating the diversity of extracellular vesicles as a whole.

The manuscript is well written and clearly, without the need of suggestions of a methodical nature especially considering the scientific rigor shown already in the introduction, line 55, in which the authors referred to the guidelines of the international society for extracellular vesicles (just a reminder, it is “for” extracellular vesicles and not “of”).

Anyway, I have some small suggestions

Line 41: lymphoid instead lymphocytic
Line 186: add a reference in support of CD34 as a marker for leukemia-derived EVs.
Line 464: For intellectual honesty, I suggest adding "could" before constitute. The results of this manuscript may be promising, but they still need to be validated by more scholars in the scientific community, so we need to be careful with overstatements.
Between line 73 and 74, a brief passage on the miRNAs contained in leukemia-derived vesicles could be added, given the international prominence, with some newer references (PMID: 32937811)

Nice job! 

Author Response

The manuscript is well written and clearly, without the need of suggestions of a methodical nature especially considering the scientific rigor shown already in the introduction, line 55, in which the authors referred to the guidelines of the international society for extracellular vesicles (just a reminder, it is “for” extracellular vesicles and not “of”).

We thank the reviewer for his positive and generous remarks.

We apologize for the mistake made in the name of ISEV- we have now modified it accordingly.

Anyway, I have some small suggestions

Line 41: lymphoid instead lymphocytic

We modified now the text in the revised version of the manuscript as suggested (now line 45).

Line 186: add a reference in support of CD34 as a marker for leukemia-derived EVs.

In support of CD34 as a marker for leukemia-derived EVs, we added the following reference: Szczepanski, M.J., et al., Blast-derived microvesicles in sera from patients with acute myeloid leukemia suppress natural killer cell function via membrane-associated transforming growth factor-beta1. Haematologica, 2011. 96(9): p. 1302-9.

Line 464: For intellectual honesty, I suggest adding "could" before constitute. The results of this manuscript may be promising, but they still need to be validated by more scholars in the scientific community, so we need to be careful with overstatements.

In the Conclusion section of the revised manuscript, we modified the sentence as suggested by the reviewer: “These antigens could constitute potential new targets for therapy and new biomarkers of the disease (now line 595).”

Between lines 73 and 74, a brief passage on the miRNAs contained in leukemia-derived vesicles could be added, given the international prominence, with some newer references (PMID: 32937811)

In the revised version of the manuscript, we added a sentence as suggested by the reviewer: “In particular, miRNAs have been shown to be transported by tumor-derived EVs in various hematologic diseases and to modulate gene expression in surrounding target cells, as in CLL [2] and multiple myeloma [3, 4]. (Line 71-74).”

Round 2

Reviewer 3 Report

I am satisfied with the author's response. I only suggest to the authors to clarify within the text, that this is a pilot study, thus to justify the sample size